# Transition to Fast Whole-Body SPECT/CT Bone Imaging: An Assessment of Image Quality

**DOI:** 10.3390/diagnostics12122938

**Published:** 2022-11-24

**Authors:** Mansour Alqahtani, Kathy Willowson, Roger Fulton, Chris Constable, Peter Kench

**Affiliations:** 1Discipline of Medical Imaging Science, Faculty of Medicine and Health, The University of Sydney, Camperdown, NSW 2006, Australia; 2Department of Radiology and Medical Imaging, Faculty of Applied Medical Sciences, Najran University, Najran 61441, Saudi Arabia; 3Department of Nuclear Medicine, Royal North Shore Hospital, St Leonards, NSW 2065, Australia; 4Institute of Medical Physics, Faculty of Science, The University of Sydney, Sydney, NSW 2006, Australia; 5Department of Medical Physics, Westmead Hospital, Sydney, NSW 2145, Australia; 6HERMES Medical Solutions, Strandbergsgatan 16, 112 51 Stockholm, Sweden

**Keywords:** SPECT/CT, bone imaging, acquisition time, image reconstruction, resolution recovery

## Abstract

Objective: To investigate the impact of reduced SPECT acquisition time on reconstructed image quality for diagnostic purposes. Method: Data from five patients referred for a routine bone SPECT/CT using the standard multi-bed SPECT/CT protocol were reviewed. The acquisition time was 900 s using gating technique; SPECT date was resampled into reduced data sets of 480 s, 450 s, 360 s and 180 s acquisition duration per bed position. Each acquisition time was reconstructed using a fixed number of subsets (8 subsets) and 4, 8, 12, and 16 iterations, followed by a post-reconstruction 3D Gaussian filter of 8 mm FWHM. Two Nuclear Medicine physicians analysed all images independently to score image quality, noise and diagnostic confidence based on a pre-defined 4-point scale. Results: Our result showed that the most frequently selected categories for 480 s and 450 s images were good image quality, average noise and fair confidence, particularly at lower iteration numbers 4 and 8. For the shortened acquisition time of 360 s and 180 s, statistical significance was observed in most reconstructed images compared with 900 s. Conclusion: The SPECT/CT can significantly shorten the acquisition time with maintained image quality, noise and diagnostic confidence. Therefore, acquiring data over 480 s and 450 s is feasible for WB-SPECT/CT bone scans to provide an optimal balance between acquisition time and image quality.

## 1. Introduction

Planar bone scintigraphy (PBS) is commonly utilised to evaluate bone disease in cancer patients due to its availability, high sensitivity, and ability to scan the entire body. Adding single photon emission tomography/computed tomography (SPECT/CT) as an adjunct to the PBS improves anatomical localisation and morphological evaluation of the equivocal lesion [1,2]. Technological innovations in SPECT/CT hardware and iterative image reconstruction with attenuation and scatter correction, including resolution recovery, have been the most significant developments in recent years, allowing for enhancement of the quality of nuclear medicine bone imaging [3,4,5]. These advancements in SPECT/CT devices have made them more favourable than SPECT-only, and widespread adoption has strengthened the position of SPECT/CT.

Recently, the novel approach of whole-body (WB) SPECT/CT has emerged, that entails the acquisition of successive axial fields of view (FOV), also called bed positions [6] to cover a longer axial range of the body than the typical 38–40 cm with single FOV SPECT/CT. Clinical evidence suggests that WB-SPECT/CT is a helpful tool for evaluating bone metastases (BM), because it reduces equivocal findings and displays effective performance in contrast to PBS [7,8,9]. Consequently, WB-SPECT/CT may substitute PBS entirely [10,11]. However, current standard SPECT/CT acquisition time, such as that used when performing single FOV SPECT/CT, is usually around 15 min or more [12,13]. This could make the total study time prohibitively long if applied directly to WB-SPECT/CT, which can require three or more axial FOV [10].

Standard bone SPECT/CT protocols, if applied to WB-SPECT/CT, would give an impractically long total acquisition time. This adversely affects patient’s comfort resulting in motion during imaging. Reduction of SPECT/CT image time would therefore aid the introduction of WB-SPECT/CT into clinical routine by reducing the likelihood of patient motion as well as bringing the total study acquisition time down to approximately the same as current PBS protocols. In recent studies, fast SPECT protocols have shown promise on reducing imaging time without loss of diagnostic quality [14,15]. In addition, it is possible to substitute “true” PBS with re-projected “synthetic” PBS derived from the WB-SPECT/CT, as has been done previously with lung imaging [16,17]. Such a workflow allows planar images to be available to aid in clinician reading, which is preferable during a transition phase to tomographic only data due to historical familiarity, without the requirement for additional scanning. Therefore, this study aims to investigate the impact of reduced SPECT/CT acquisition time on reconstructed image quality. 

## 2. Materials and Methods

### 2.1. Patient Selection

Approval for the study was obtained from the Northern Sydney Local Health District Human Research Ethics Committee (Ref 02150/2020). This retrospective study recruited 5 patients who were referred for a routine ^99m^Tc-hydroxymethylene diphosphonate (^99m^Tc-HDP) bone SPECT/CT. The clinical indications included a variety of oncological, infection and/or chronic diseases. The study was performed in the Nuclear Medicine Department, Royal North Shore Hospital, Sydney, Australia. Of the five patients included in the study, the clinical area of interest for two was lumbar/thoracic spine, for another two it was knees, and for one the Pelvis. The age range of the patients was 44–89 years. The Patient characteristics are shown in Table 1.

### 2.2. Data Acquisition and Image Gating

The patients received an intravenous injection of 863.6 ± 20 (mean ± SD) MBq of ^99m^Tc-HDP. All SPECT data were acquired 2–3 h after injection on a Symbia T16 dual-detector SPECT/CT scanner (Symbia T16, Siemens Healthineers, USA) equipped with low-energy high-resolution (LEHR) collimators. Data were acquired in opposing configuration with a 128 × 128 matrix size, 4.8 mm pixels, 15 s per projection over 360 degrees for a total of 120 projections, in a non-circular contoured orbit and step-and-shoot mode. A low-dose CT scan was obtained for attenuation correction and anatomical localization. CT data were acquired with exposure of 130 kV, 10 effective mAs, pitch of 1.5, employing adaptive dose modulation (CARE Dose 4D, Siemens Medical Solutions) and reconstructed into 4.8 mm-thick slices using a smooth reconstruction kernel B31s Siemens Healthcare [18]. 

To avoid the need to re-image patients for different acquisition durations, resampling of the standard acquired SPECT data into equal time bins was instead utilised. It is possible to explore the impact of decreased acquisition times by using image gating, which is supported by most modern SPECT cameras. This technique has been demonstrated by Bailey and Kalemis [19], using an electrocardiography (ECG) simulator as an external trigger to generate non-physiological gating, resulting in essentially identical yet statistically distinct partitioned datasets at reduced acquisition time. A more in depth discussion of this technique can be found in our previous work [14]. 

In our study, the original acquisition time was 15 s per projection angle. We applied acquisition gating into 15-time bins of equal duration, therefore each gated time bin corresponded to 1 s of acquisition time. This allowed for the generation of ungated data of any duration, from 1 s to 15 s per projection, by simple addition of the time bins (Figure 1). Projection data summing was done using in-house software written in Interactive Data Language Program (Research Systems International, Boulder, CO, USA). Using this technique, data corresponding to projection times of 8, 6, and 3 s could be derived for comparison to the full data set acquisition time of 15 s per projection. 

In addition, a second set of raw SPECT projection data was generated with only 60 projections angles, instead of 120 angles, with 15 s duration per projection, thereby reducing the total acquisition time by half. This was achieved by discarding every other projection from the raw data and saving a new dataset.

### 2.3. Image Reconstruction

The SPECT acquisition data were sent back to the Siemens system for reconstruction after being summed to different acquisition times. All images in this study were reconstructed using a fixed number of subsets (8 subsets) and 4, 8, 12, and 16 iterations, followed by a post-reconstruction 3D Gaussian filter of 8 mm FWHM. For every patient the following datassets were analysed (Figure 1): (1)Reference: full acquisition time (15 s per projection) reconstructed with a total of 120 views (60 stops) resulted in a total acquisition time of 900 s.(2)Reduced acquisition of 8, 6, and 3 s per projection angle with a total of 120 views (60 stops) resulted in a total acquisition time of 480, 360 and 180 s, respectively.(3)Full acquisition duration (15 s per projection) with a total of 60 projections (30 stops) resulted in a total acquisition time of 450 s.

SPECT data of varying acquisition times were reconstructed using the 3D Ordered Subset Expectation Maximization (OSEM) algorithm (Siemens Healthiness, USA). This algorithm applies resolution recovery (RR) using a distance-dependent 3D Gaussian collimator-detector model, attenuation correction (AC) based on a CT-derived linear attenuation map, and scatter correction (SC) using scatter window subtraction (dual energy window approach).

This resulted in a total of 100 images for evaluation, 20 for each patient. For the sake of simplicity, the equivalent total acquisition time for the five acquisition protocol strategies using 15 s-120 projections (reference data), 8 s-120 projections, 15 s-60 projections, 6 s-120 projections, and 3 s-120 projections is referred to as 900 s, 480 s, 450 s, 360 s, and 180 s, respectively, in the rest of this paper.

### 2.4. Clinical Image Quality Assessment

All 100 clinical studies were reconstructed, and then the images were divided into five batches (20 images/batch) for comparison. Each batch corresponds to only one acquisition time but contained all five patients and all four reconstructions. The images were presented in random order. Two senior NM specialists’ physicians were engaged to read the clinical images. Each data set was anonymised such that the readers were blinded to patient identifiers, acquisition time and the reconstruction method applied. Using a 4-point scale (Table 2), the readers independently evaluated the image quality, image noise and diagnostic confidence of the SPECT/CT data. Data sets were viewed using multiplanar (transverse, coronal, and sagittal) reformatted sections. The reading sessions were separated by days to minimise the possibility of remembering the presented images from the previous session. All clinical images were displayed to the readers using medical image viewers by MIM (MIM Software Inc, Cleveland, OH, USA) and Syngo.Via, version VB10 (Siemens Healthcare Software, Erlangen, Germany). An example of a reader’s subjective image quality grading is demonstrated in Figure 2.

### 2.5. Statistical Analysis

Due to the non-parametric nature of the data, the Kruskal–Wallis test was performed to test for statistically significant variations in the image evaluation scores between all five different acquisition times. Following the Kruskal–Wallis test, a post hoc multiple comparison test was also used to see where the major variations were [20]. A *p* value less than 0.05 was considered significant. All statistical evaluations were performed with GraphPad Prism 9.0 Software (GraphPad Software, San Diego, CA, USA).

## 3. Results

### 3.1. Clinical Image Quality Assessment

The subjective image quality was evaluated using a 4-point scale covering three aspects: image quality, image noise and diagnostic confidence. The image quality, image noise and diagnostic confidence scores for all images grouped by acquisition times of 900 s, 480 s, 450 s, 380 s and 180 s were compared and were found to be significantly different (all *p* < 0.001). 

Table 3, Table 4 and Table 5 represents the results when comparing image quality scores, image noise scores, and diagnostic confidence scores for clinical data of varying acquisition duration (900 s, 480 s, 450 s, 360 s, 180 s), all reconstructed with 4, 8, 12 and 16 iterations, 8-subsets and 8 mm FWHM 3D Gaussian filter.

#### 3.1.1. Image Quality

Table 3 shows the mean and standard deviation of image quality scores on a 4-point Likert scale among two readers at varied acquisition times. Different acquisition times of 900 s, 480 s, 450 s, 360 s and 180 s with 4, 8, 12, and 16 iterations were observed. It was discovered that the number of iterations and the score value were inversely proportional. With 4 and 8 iteration numbers, the 480 s scored higher and had superior image quality than the other two shorter time acquisitions of 360 s and 180 s (all *p* < 0.05). In contrast, there was no significant difference between 900 s and 480 s, and there was also no big variation among 480 s and 450 s (Appendix A). However, the scoring results showed that 480 s had a superior grading level than 450 s (full acquisition timer per projection but lower number of projections), particularly at lower iteration numbers 4 and 8.

#### 3.1.2. Image Noise

The image noise scores for 900 s, 480 s_,_ 450 s_,_ 360 s and 180 s with 4, 8, 12, and 16 iterations, are shown in Table 4. There was a reduction in noise with increased acquisition time. With 4 and 8 iteration numbers, 480 s and 450 s showed improvement and had less noise than the other two shorter time acquisitions of 360 s and 180 s. However, at 4 iterations, there was a significant difference between 900 s and 480 s (*p* = 0. 012) and between 900 s and 450 s (*p* = 0. 004). In addition, significant differences were observed between 900 s and 360 s and between 900 s and 180 s regardless of the number of iterations, focusing on the image noise response analysis for the readers (Appendix A).

The reader’s perception of noise increased as the number of iterations increased (4, 8, 12, and 16 iterations). Therefore, the noise level at a lower iteration number of 4 and 8 yielded higher scores value with average noise score category for 480 s and 450 s. On the other hand, the noise grading level with very low acquisition times of 360 s and 180 s ranged between moderate and excessive noise categories.

#### 3.1.3. Diagnostics Confidence

Table 5 shows diagnostic confidence scores on a 4-point Likert scale for two readers at varied acquisition times. Diagnostic confidence ratings of 900 s 480 s and 450 s were considerably higher than those with shorter acquisition times. In contrast, there was no significant difference between reference images 900 s and 480 s (all *p* > 0.12), and there was also no significant variation between reference 900 s images and 450 s (all *p* > 0.45) (Appendix A).

The mean and standard deviation scores of combined iteration number (20 images) evaluated by two readers for image quality, image noise, and diagnostic confidence is shown in Figure 3. When comparing the grading between 900 s and 480 s and between 480 s and 450 s_,_ there were only minimal differences in all three quality responses. To further our understanding of the significance of the change between the number of iterations, the result was evaluated individually for each reader (Appendix A). In general, both readers were consistent in their grading between different iteration numbers. Therefore, there was no statistically significant difference between data reconstructed with varying iterations for a given acquisition duration.

## 4. Discussion

This study evaluated different acquisition durations and iterative reconstruction parameters by specialists’ review with the ultimate objective of determining the shortest acquisition time for diagnostic WB bone-SPECT/CT without significantly compromising image quality, noise and diagnostic confidence. We used a standard multi-bed bone SPECT/CT protocol as a reference. Overall, the experimental scheme for image quality, noise and diagnostic confidence resulted in significant differences between very low acquisition times of 360 s and 180 s with 900 s (*p* < 0.05) with the exception of the diagnostic confidence at 8 iterations (*p* = 0.11) and image quality, noise and diagnostic confidence at 16 iteration (all *p* > 0.05). In addition, the most frequently selected categories for 480 s and 450 s images were good image quality, average noise and fairly confidence, particularly at lower iteration numbers 4 and 8, with no statistically significant difference between the two acquisition times. Furthermore, the findings in the current study showed that reducing scanning time by approximately 50% (when compared to the current clinical protocol) did not significantly influence image quality, noise or diagnostic confidence. Therefore, using 480 s and 450 s with a low number of iteration (4–8) could be feasible for WB-SPECT/CT bone scan. 

As expected, compared to the reference images-900 s, the variant increased with a lower acquisition time of 360 s and 180 s. Readers 1 and 2 scored a decrease in image quality level with reduced acquisition time for images reconstructed using 360 s and 180 s with different iteration numbers, compared to 900 s, with an average of 35% and 48% for reader 1 and 30% and 49% for reader 2, respectively (Figure 3). On the other hand, when the acquisition time is reduced to 480 s, the image quality score slightly decreases with an average of 11% for both readers. In contrast, when only half the projections are acquired at full frame duration, resulting in a total acquisition time of 450 s, the first reader scored lower than the second reader, with an average of 18%. However, the second reader gave 480 s and 450 s almost similar average values at 11%, yet there was no significant difference between readers.

The increased noise level was observed by both readers for the images with very low acquisition times of 360 s and 180 s, compared to the 900 s, with an average of 31% and 47% for reader 1 and 40% and 56% for reader 2, respectively. On the contrary, when the acquisition duration is reduced to 480 s, the average image noise score reduces by 14% and 21% for readers 1 and 2, respectively (Figure 3). However, there were no significant variations across readers. The diagnostic confidence level decreased with an average of 13% and 9% at 480 s for reader 1 and reader 2, respectively, and only a slightly lower level of an average of 12% and 8% for 450 s for reader 1 and reader 2, respectively, when compared to the 900 s.

Although there was no statistically significant difference between data reconstructed with varying numbers of iterations for a given acquisition duration (Appendix A), there was a substantial difference observed between low acquisition times of 360 s and 180 s and the reference images-900 s. Low statistical counts of 360 s and 180 s indicate an increase in noise levels, hence degrading image quality and diagnostic confidence. In contrast, longer acquisition times yield more clear and diagnostic images (Figure 4). 

Consistent with our result, a previous study [14] showed the sphere’s deformation became more apparent with the short acquisition time using NEMA IEC Body phantom with irregularly shaped lesion simulating inserts. In our study, the high noise induced by low counts in images at 360 s and 180 s was the principal cause of poor image quality. A previous study have shown that “active bone lesions will be visible even at very low scan duration, although this may not be the case for lesions with a lower uptake or degenerative changing diseases” [14]. Therefore, SPECT/CT images produced with a short acquisition duration may have difficulties detecting these lesions. In addition, in the current study the NM specialists prefer the lower number of iterations because of the noise amplification increased with increasing iteration number (Figure 5). However, it should be considered that the physician reader preferences also play a part in determining the optimal balance of image reconstruction against noise levels when acquisition time is reduced.

In the present study, all images were reconstructed with RR. Several studies have highlighted the potential of utilising the RR method to enhance SPECT image quality during iterative reconstruction. For example, Thientunyakit et al. [21] evaluated the diagnostic performance of multiplanar pelvic bone scan and half-time SPECT in patients diagnosed with severe bladder artefacts. They reported that the diagnostic certainty with which equivocal pelvic lesions were interpreted significantly improved in comparison to multiplanar imaging; additional advantages were reported, including that the staff were exposed to lower radiation dosage and the scan duration was reduced. Borges-Neto et al. [22] conducted phantom and clinical investigations and demonstrated that RR can generate comparable image quality with lower scan periods. Aldridge et al. [23] study of image quality in half-time imaging in parathyroid and bone SPECT/CT is consistent with our findings. They discovered that using 3D RR for image reconstruction reduced acquisition time by 10–15 min without compromising image quality.

A recent study by Kapsoritakis et al. started implementing WB-SPECT/CT protocol and omitting PBS in their routine practice with the high suspicion of metastatic disease and obese patients. In this study, their findings showed that WB-SPECT/CT had a diagnostic superiority over PBS. As a result, they recommended the WB-SPECT/CT protocol for all oncology patients [24]. However, the process was time-consuming as the acquisition time by 15–20 min per bed position. By shortening the SPECT acquisition time duration, WB-SPECT/CT can be introduced into clinical practice. Our findings showed a significant reduction for the acquisition time to 7.5–8 min per bed position with maintaining image quality, noise and diagnostic confidence. 

The primary limitation of this study was the single site clinical setting and small sample size. However, despite the relatively small number of patients, we have demonstrated a significant difference between images in different acquisition durations with scoring trends that remained consistent across all studies. In addition, only two readers participated in the evaluation of image quality, noise and diagnostic confidence. This study laid the foundation for further large-scale research to explore the same experiments in multiple centers using various types of dual head SPECT/CT scanners and with the participation of numerous readers.

## 5. Conclusions

Overall, the current study demonstrated that reducing scanning time by approximately 50% of the typical clinical imaging protocol for bone SPECT/CT did not significantly influence image quality, noise and diagnostic confidence as scored by two experienced NM physicians. Therefore, we propose using 480 s or 450 s and low iterative OSEM reconstruction (e.g., 4–8 iterations, including AC, SC and RR) as a feasible approach to WB-SPECT/CT bone studies to maintain diagnostic confidence in line with patient comfort and clinical throughput.

## Figures and Tables

**Figure 1 diagnostics-12-02938-f001:**
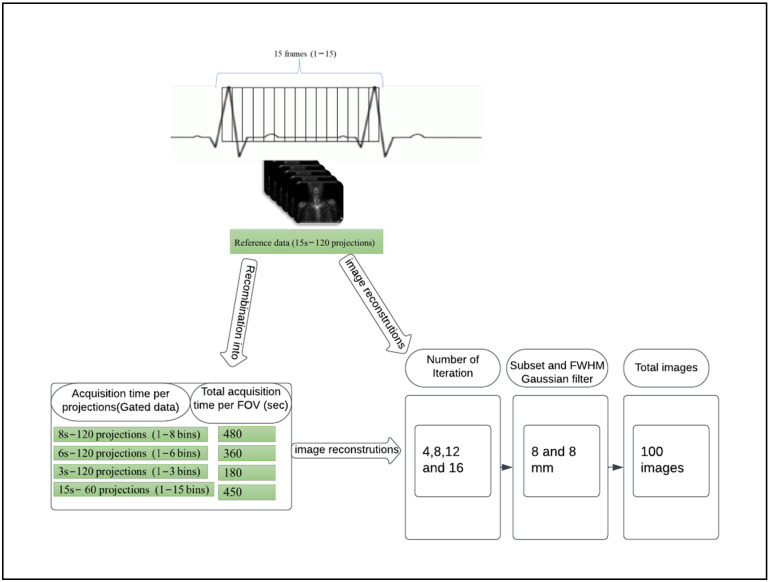
Process for partitioning and recombining bins into lower acquisition time and the image reconstructions utilised at each acquisition period.

**Figure 2 diagnostics-12-02938-f002:**
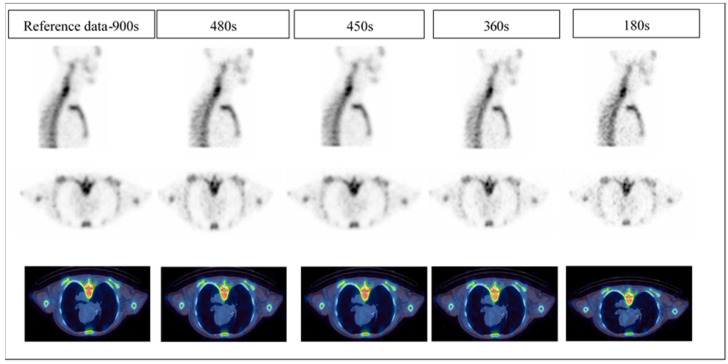
An example of subjective image quality. The image quality scores of 4, 3, 3, 2 and 1 were given to the reference images-900 s, 480 s, 450 s, 360 s and 180 s, respectively.

**Figure 3 diagnostics-12-02938-f003:**
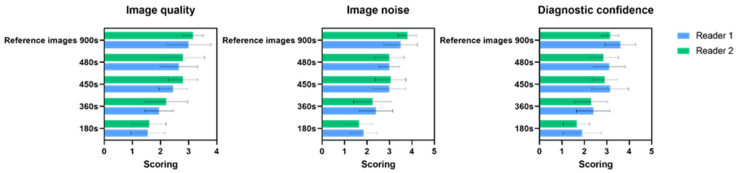
Bar plots showing the mean values and standard deviations of the clinical readings for each reader at different acquisition protocol strategies for the three-quality assessment (image quality, image noise and diagnostic confidence.

**Figure 4 diagnostics-12-02938-f004:**
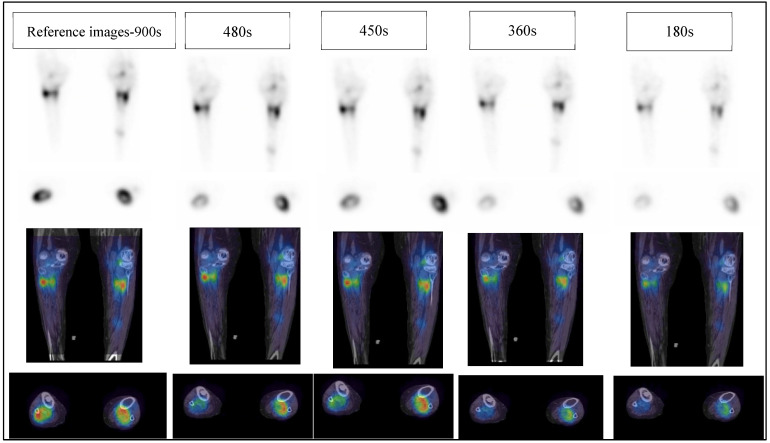
The top two rows show SPECT images, and the bottom two rows show SPECT/CT images from left to right reference images-900 s, 480 s, 450 s, 360 s and 180 s, respectively. All reconstructions used 4 iterations, 8 subsets, and an 8 mm Gaussian filter.

**Figure 5 diagnostics-12-02938-f005:**
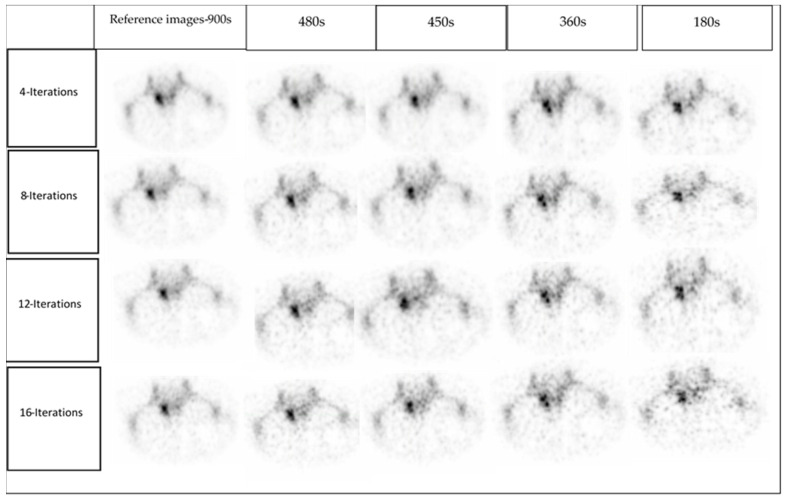
Example for SPECT images at different iteration number in different acquisition time. This patient had Osteoplastic activity in the left Lumbar 1/sacrum 1.

**Table 1 diagnostics-12-02938-t001:** Patients’ characteristics.

Patient	Gender	Age (Years)	Tracer	Injected Dose (MBq)	Diagnosis
1	F	44	^99m^Tc HDP	841	Cervical spine spondylodiscitis
2	M	55	^99m^Tc HDP	840	Degenerative changes with mild moderate Osteoplastic activity in cervical spine.
3	F	76	^99m^Tc HDP	881	Multiple fractures in both tibiae and right fibular head
4	M	89	^99m^Tc HDP	892	Osteoplastic activity in the left Lumber 5/sacrum 1 with degenerative change in the hip joints.
5	M	60	^99m^Tc HDP	864	Osteoplastic activity with degenerative changes

**Table 2 diagnostics-12-02938-t002:** 4-point Likert scale for subjective SPECT/CT image quality.

	Image Quality	Image Noise	Diagnostic Confidence
1	Unacceptable	Excessive noise	Not confident
2	Acceptable	Moderately increased	Slightly confident
3	Good	Average noise	Fairly confident
4	Excellent	Low image noise	Highly confident

**Table 3 diagnostics-12-02938-t003:** Image quality scores by 4-point Likert scale across 2 readers for different acquisition times.

Iterations Number	Reference Images-900 s	480 s		450 s		360 s		180 s		All
	Mean ± SD	Mean ± SD	*p* Values ^(1)^	Mean ± SD	*p* Values ^(1)^	Mean ± SD	*p* Values ^(1)^	Mean ± SD	*p* Values ^(1)^	*p* Values ^(2)^
4	3.50 ± 0.52	3.10 ± 0.73	0.79	2.80 ± 0.63	0.097	2.30 ± 0.82	0.011 *	1.70 ± 82	<0.001 *	<0.001 *
8	3.20 ± 0.63	2.90 ± 0.73	0.86	2.60 ± 0.51	0.18	2.00 ± 0.47	0.001 *	1.50 ± 0.52	<0.001 *	<0.001 *
12	2.90 ± 0.31	2.40 ± 0.69	0.18	2.60 ± 0.51	0.55	2.00 ± 0.47	0.005 *	1.50 ± 0.52	0.001 *	<0.001 *
16	2.70 ± 0.67	2.50 ± 0.52	0.73	2.50 ± 0.52	0.73	2.00 ± 0.81	0.054	1.60 ± 0.51	0.014 *	0.004 *

The mean value ± SD represents the mean score of 5 patient’s images, and two readers scored each image. *p* values * ^(1)^; Significance value for each acquisition time compared to the reference images 900 s (all reconstructed with identical parameters). *p* values * ^(2)^; Significance value for all five acquisition times group.

**Table 4 diagnostics-12-02938-t004:** Image noise scores by 4-point Likert scale across 2 readers for different acquisition times.

Iterations Number	Reference Images-900 s	480 s		450 s		360 s		180 s		All
	Mean ± SD	Mean ± SD	*p* Values ^(1)^	Mean ± SD	*p* Values ^(1)^	Mean ± SD	*p* Values ^(1)^	Mean ± SD	*p* Values ^(1)^	*p* Values ^(2)^
4	4.00 ± 0.00	3.30 ± 0.48	0. 012 *	3.10 ± 0.56	0.004 *	2.30 ± 0.82	<0.001 *	2.00 ± 0.81	<0.001 *	<0.001 *
8	3.60 ± 0.69	3.10 ± 0.56	0.429	3.15 ± 0.73	0.542	2.40 ± 0.84	0.021 *	1.60 ± 0.51	<0.001 *	<0.001 *
12	3.60 ± 0.69	2.90 ± 0.56	0.131	3.10 ± 0.87	0.638	2.40 ± 0.69	0.021 *	1.70 ± 0.48	0.002 *	<0.001 *
16	3.40 ± 0.69	2.70 ± 0.48	0.143	2.80 ± 0.63	0.320	2.20 ± 0.91	0.051	1.70 ± 0.48	0.002 *	<0.001 *

The mean value ± SD represents the mean score of 5 patient’s images, and two readers scored each image. *p* values * ^(1)^; Significance value for each acquisition time compared to the reference images 900 s (all reconstructed with identical parameters). *p* values * ^(2)^; Significance value for all five acquisition times group.

**Table 5 diagnostics-12-02938-t005:** Diagnostic confidence scores by 4-point Likert scale across 2 readers for different acquisition times.

Iterations Number	Reference Images-900 s	480 s		450 s		360 s		180 s		All
	Mean ± SD	Mean ± SD	*p* Values ^(1)^	Mean ± SD	*p* Values ^(1)^	Mean ± SD	*p* Values ^(1)^	Mean ± SD	*p* Values ^(1)^	*p* Values ^(2)^
4	3.70 ± 0.48	3.10 ± 0.56	0.125	3.20 ± 0.78	0.458	2.50 ± 0.70	0.003 *	1.90 ± 0.87	<0.001 *	<0.001 *
8	3.30 ± 0.67	3.10 ± 0.73	0.968	3.00 ± 0.47	0.777	2.50 ± 0.70	0.115	1.60 ± 0.51	<0.001 *	<0.001 *
12	3.40 ± 0.51	2.90 ± 0.87	0.546	3.00 ± 0.81	0.690	2.30 ± 0.67	0.006 *	1.80 ± 0.91	0.002 *	<0.001 *
16	3.10 ± 0.56	2.80 ± 0.63	0.798	2.90 ± 0.73	0.959	2.10 ± 0.87	0.082	1.80 ± 0.63	0.006 *	<0.001 *

The mean value ± SD represents the mean score of 5 patient’s images, and two readers scored each image. *p* values * ^(1)^; Significance value for each acquisition time compared to the reference images 900 s (all reconstructed with identical parameters). *p* values * ^(2)^; Significance value for all five acquisition times group.

## Data Availability

Data can be made available upon reasonable request to the corresponding author. The data are not publicly available owing to privacy.

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
