# Peer review of "Transition to Fast Whole-Body SPECT/CT Bone Imaging: An Assessment of Image Quality"

_diagnostics, 2022, doi:10.3390/diagnostics12122938_

Round 1
Reviewer 1 Report
This is a well-performed study on WB-SPECT/CT for patients undergoing screening for bone disease. Although I disagree with the authors that typical planar images will be replaced by WB-SPECT/CT given time, costs, availability of personnel, patient tolerability, and increased radiation exposure with questionable outcome benefits, I think the study has merit by itself. Of course, it is severely limited by the low sample size, but I see this more as a case series than anything. It is also unclear how authors talked people who were referred for focal imaging to undergo full-body SPECT/CT, but it does not change the results/conclusion. I think the paper reads well as is limitations acknowledged.
Author Response
Dear Editors,
Thank you for giving us the opportunity to submit a revised draft of the manuscript “Transition to fast whole-body SPECT/CT bone imaging: an assessment of image quality” for publication in the diagnostic journal. We appreciate the time and effort you and the reviewers have dedicated to providing feedback on our manuscript.
We have incorporated the suggestions made by the reviewer. Those changes are highlighted within the manuscript in red. Please see below, in blue, for a point-by-point response to the reviewers.
Comments from Associate Editor (Remarks to the Author)
This is a well-performed study on WB-SPECT/CT for patients undergoing screening for bone disease. Although I disagree with the authors that typical planar images will be replaced by WB-SPECT/CT given time, costs, availability of personnel, patient tolerability, and increased radiation exposure with questionable outcome benefits, I think the study has merit by itself. Of course, it is severely limited by the low sample size, but I see this more as a case series than anything.
Response: We appreciate your feedback and interesting comments. There relatively few published studies that deal directly with this question. We do not state that planar bone scintigraphy “will be replaced” by WB-SPECT/CT however we do have evidence that they could be and that it might even be likely in the near future.
Recently, several studies have proposed that WB-SPECT/CT could replace PBS in in the near future (1, 2). Also, some next-generation gamma cameras recently introduced are SPECT-only systems that could enable WB tomographic imaging in a short acquisition time (3). Therefore, the transition from PBS to a routine WB-SPECT/CT workflow seems probable in the near future (4). This technique allows 2D images to still be available to aid in clinician reading, which is thought to be preferable during a transition phase to 3D only data due to historical familiarity, whilst saving acquisition time. Such a review is currently in progress.
- Abikhzer G, Gourevich K, Kagna O, Israel O, Frenkel A, Keidar Z. Whole-body bone SPECT in breast cancer patients: the future bone scan protocol? Nucl Med Commun. 2016;37(3):247-53.
- Palmedo H, Marx C, Ebert A, Kreft B, Ko Y, Türler A, et al. Whole-body SPECT/CT for bone scintigraphy: diagnostic value and effect on patient management in oncological patients. Eur J Nucl Med Mol Imaging. 2014;41(1):59-67.
- Huh Y, Yang J, Dim OU, Cui Y, Tao W, Huang Q, et al. Evaluation of a variable‐aperture full‐ring SPECT system using large‐area pixelated CZT modules: A simulation study for brain SPECT applications. Med Phys. 2021;48(5):2301-14.
- Melki S, Chawki MB, Marie P-Y, Imbert L, Verger A. Augmented planar bone scintigraphy obtained from a whole-body SPECT recording of less than 20 min with a high-sensitivity 360 CZT camera. Eur J Nucl Med Mol Imaging. 2020;47(5):1329-31.
It is also unclear how authors talked people who were referred for focal imaging to undergo full-body SPECT/CT, but it does not change the results/conclusion.
Response: Thank you for your comment. We added the paragraphs below to the method section.
The patients who were included in this study were referred for Whole Body Bone imaging. The clinical indications included a variety of oncological, infection and/or chronic diseases for which whole body imaging was deemed appropriate. We have added this description to the manuscript for clarity.
Reviewer 2 Report
The paper under evaluation is an interesting prospective study assessing the impact of different acquisition durations and iterative recontruction parameters on image quality when WB-SPECT/CT is employed.
It is a very good paper, clearly presented, although leaning more towards the technological aspects rather than on the diagnostic ones. The authors suggest that a reduction of 50% in scanning time might be applied without significantly hampering image quality.
Limitations are clearly stated in the manuscript. Images are illustrative.
Here a suggestion:
- correctly the authors investigated the feasibility of a short protocol in bone scan, but I think that it might be of value, for example, for WB-SPECT/CT in case of infection and while blood cell scintigraphy. Did the authors have some thoughts about this potential application? A brief paragraph would be of value.
Author Response
Dear Editors,
Thank you for giving us the opportunity to submit a revised draft of the manuscript “Transition to fast whole-body SPECT/CT bone imaging: an assessment of image quality” for publication in the diagnostic journal. We appreciate the time and effort you and the reviewers have dedicated to providing feedback on our manuscript.
We have incorporated the suggestions made by the reviewer. Please see below, in blue, for a point-by-point response to the reviewers.
Comments from Associate Editor (Remarks to the Author)
The paper under evaluation is an interesting prospective study assessing the impact of different acquisition durations and iterative recontruction parameters on image quality when WB-SPECT/CT is employed.
It is a very good paper, clearly presented, although leaning more towards the technological aspects rather than on the diagnostic ones. The authors suggest that a reduction of 50% in scanning time might be applied without significantly hampering image quality.
Limitations are clearly stated in the manuscript. Images are illustrative.
Response: We appreciate your positive feedback and interesting comments.
Here a suggestion:
Correctly the authors investigated the feasibility of a short protocol in bone scan, but I think that it might be of value, for example, for WB-SPECT/CT in case of infection and while blood cell scintigraphy. Did the authors have some thoughts about this potential application? A brief paragraph would be of value.
Response: This is a good question. Our response is that we have only studied 99mTc Bone imaging. The results are probably transferable to other 99mTc imaging scenarios, such as 99mTc labelled WBC imaging, or 99mTc PSMA. Whether they are applicable to other isotopes (e.g. 67Ga, 111In) is perhaps less certain. In any case they clinical imaging protocol should be validated for each application. This includes the reconstruction as well as the acquisition protocols.